# The Role of Family Resilience and Pandemic Burnout on Mental Health: A Two-Wave Study in China

**DOI:** 10.3390/ijerph20053803

**Published:** 2023-02-21

**Authors:** Catherine So-kum Tang, Tiffany Sok U Siu, Tak Sang Chow, Helen Sin-Hang Kwok

**Affiliations:** 1Department of Counselling and Psychology, Hong Kong Shue Yan University, North Point, Hong Kong SAR, China; 2Centre for Interdisciplinary Evidence-based Practice and Research, Hong Kong Shue Yan University, North Point, Hong Kong SAR, China

**Keywords:** family resilience, pandemic burnout, anxiety and depression, longitudinal study, COVID-19

## Abstract

Family resilience refers to the processes through which a family adapts to and bounces back from adversities. Pandemic burnout refers to feeling emotionally exhausted, cynical, and lack of accomplishment during the pandemic and/or toward various preventive polices and measures. This two-wave, region-wide, longitudinal study included 796 adult participants residing in mainland China. Participants completed online surveys at two time points during the COVID-19 pandemic. Time 1 (T1) survey was conducted when the number of new infected cases in China stabilized, while Time 2 (T2) was conducted 5 months later when there was a sudden surge of new infected cases. Results of a hierarchical regression analysis revealed that the interaction and main effects of pandemic burnout and family resilience at T2 showed significant incremental prediction of depression and anxiety at T2, after controlling for demographic as well as individual and family resilience at T1. These results supported the hypotheses that current family resilience functions as a protective factor, whereas pandemic burnout functions as a risk factor of mental health during successive waves of pandemic outbreaks. In particular, family resilience at T2 mitigated the negative impact of high pandemic burnout on anxiety and depression at T2.

## 1. Introduction

### 1.1. Mental Health in the Time of COVID-19 Pandemic

Since the first reported case in late December 2019, the coronavirus disease (COVID-19) soon found its way to every corner of the planet Earth. A global public health emergency was declared by the World Health Organization on the 30 January 2020 [1]. The power of the COVID-19 pandemic is far-reaching, long-lasting, and still ongoing. Consistent with our experiences, myriad and converging empirical findings of its detrimental impact on our mental health and well-being are well-documented [2,3,4,5,6]. For example, in comparison with the pre-COVID time, elevated levels of depressive symptoms [7], psychological distress [8], loneliness [9], suicidal thoughts [10], anxiety, and eating pathology [11] were found after the outbreak of COVID-19. 

Empirical investigations that compare the changes of depression and anxiety from before and during early stages of the COVID-19 pandemic outbreak have shown trends with a spike at the initial outbreak followed by a decline over time [12,13], which suggests a sign of human resilience and adaption towards the influence of COVID-19. However, there is no definite conclusion regarding how anxiety and depression manifests as the current pandemic keeps evolving. Meta-analyses that investigated the development of anxiety and depressive symptoms during COVID-19 outbreaks showed that trajectories of anxiety and depression prevalence varies among countries: some declined over time, some sustained, some went up and down [14,15]. Research has also found that the changes ebbed and flowed with the number of confirmed cases [16], intensity of lockdown implementation [17], and the risk of infecting COVID-19 in the residing area, as well as the development of COVID-19 [18]. Although some of us seemingly have gotten used to and are able to withstand the negative influences of COVID-19, it remains unclear under the chronic exposure to the ever-changing pandemic, such as discovery of new variants, on-and-off implementations of preventive measures, what would put us at a greater risk, and what would act as a protective factor towards anxiety and depression when COVID-19 has become a new normalcy.

### 1.2. Family Resilience: A Protective Factor beyond Individual Resilience

The influence of the COVID-19 pandemic is unprecedentedly monumental, not solely due to its universal reach, but also its influences on multiple systems, from an individual to the family, organization, community, and country an individual belongs to. Therefore, it should be considered a multilevel disaster, which requires a multi-systemic response [19]. This view is aligned with the theoretical development of resilience. In recent decades, there is a growing consensus that resilience should be viewed as a dynamic, multilevel, and systemic process [20], arising from the interactions between a changing individual and the contexts he or she is in when adversities emerge [21]. 

Recent scholarly work has theorized that family resilience may shield the disruptive effects of the COVID-19 pandemic [22]. Family resilience refers to the processes through which a family adapts to and bounces back from adversities. Three crucial dimensions of family resilience have been identified as belief systems, organizational patterns, and communication processes [23]. Belief systems include maintaining positive outlooks, transcendence, and meaning-making; organizational patterns consist of connectedness of family, flexibility, and use of social and economic resources; and communication processes encompass both family’s problem solving and communication patterns [24]. Indeed, it is not difficult to imagine how family resilience is protective towards COVID-19. Take belief system as an example: in the face of unprecedented adversity, such as COVID-19, families must make meaning out of it, either with their existing framework or an updated belief system. When it is perceived as interpretable, a family might see the event as manageable, which might translate into adaptive coping. 

However, few empirical endeavors have been made to examine the influence of family resilience on mental health during the COVID-19 pandemic. A cross-cultural study has explored the influence of family resilience and individual resilience simultaneously on anxiety, depression, and stress during the first wave of the COVID pandemic in 2020 [25]. It has found that both family and individual resilience had unique predictive power to anxiety and depression among Hong Kong participants, as opposite to only individual resilience was found to be negatively related to anxiety among participants in the United States. However, only two subcomponents of family resilience, namely (1) family communication and problem solving and (2) maintaining a positive outlook, were included. The third subcomponent, i.e., utilizing social resources, was excluded due to low reliability of this subscale. Other studies on family resilience also examined its subcomponents independently [24,26]. Thus, it remains unclear (1) whether, when examined holistically, family resilience processes have positive influence over anxiety and depression beyond individual resilience and (2) whether family resilience is a time-specific or relatively stable construct in relation to anxiety and depression. To address these unknowns, this study aims to examine the influence of family resilience beyond individual resilience on anxiety and depression at two time points.

### 1.3. Pandemic Burnout: An Overlooked Phenomenon and Risk Factor for Anxiety and Depression

Researchers have identified socio-demographic (e.g., gender and socioeconomic status) [27] and socio-psychological (e.g., social support, loneliness, and threat perception) [15] risk factors of anxiety and depression during the COVID-19 pandemic. Undoubtedly, these investigations are helpful in identifying vulnerable groups and informing the direction of intervention. However, at the time of writing, it has been almost three years since the WHO’s announcement of the COVID-19 pandemic as a global health emergency. COVID-19 is still affecting our everyday lives with few signs of cessation. Many environmental changes were suddenly thrust upon individuals without forewarning, including mandated quarantine, social distancing, school closure, abrupt transition to distant learning for children, remote work for parents, and financial insecurity. The successive waves of pandemic outbreaks over the last three years have thus created a “new normal” living and working style that itself also induces different types of stress and conflict within families. This may, in turn, affect the mental health of individuals and their families. In particular, individuals may experience physical, emotional, and behavioral fatigue in relation to the COVID-19 pandemic, which has been coined as “pandemic fatigue” by the national and international public sectors [28,29] and media [30]. There is also increasing research on factors such as pandemic fatigue, that may facilitate or impede individuals’ adherence to preventive measures over time [31,32,33,34]. 

An interrelated yet distinctive phenomenon related to pandemic fatigue during COVID-19 is burnout, which derives from a well-studied construct in work settings. Burnout is originally defined as a set of psychological symptoms due to prolonged exposure to interpersonal stress [35]. Similar to the three-dimensional model of burnout, pandemic burnout experience refers to feeling emotionally exhausted, cynical, and lack of accomplishment during the pandemic, under the prolonged exposure to various preventive polices and measures, and other pandemic-related stressors. Until now, the investigation of the pandemic-related burnout experience has been predominantly about burnout “during” but not “towards” the COVID-19 pandemic. Certain occupations, such as healthcare providers, are the center of burnout studies during the COVID-19 pandemic, as they have been exposed to working environments with drastically increased stress since the onset of the COVID-19 pandemic [36,37,38]. However, the general public is not completely immune from feeling burnout towards the COVID-19 pandemic, nor do they necessarily experience less pandemic burnout than healthcare providers. For example, studies have found similar levels of psychological well-being [39,40] or even worse mental health symptoms when comparing non-medical and medical professionals [41]. Currently, limited investigation has been specifically on pandemic burnout [42,43,44], without focusing on the general population but instead on healthcare providers. (For exceptions, see [32,45]). Therefore, the present study aims to examine the impact of pandemic burnout on anxiety and depression among the general public. We examine the relevance of the pandemic burnout experience on mental health when people were under stringent preventative measures at the time China had a surging number of outbreaks. 

### 1.4. Purpose and Hypothesis of the Study

A two-wave study was conducted in mainland China to investigate family resilience and pandemic burnout experience as protective and risk factors of mental health during the COVID-19 pandemic. The Time 1 survey was conducted in September 2021 when the number of new infected cases in China stabilized at below 100 per day, while Time 2 was conducted in April 2022 when there was a sudden surge of new infected cases at 1000–3000 per day.

Specific hypotheses of the study include the following:

Family resilience would function as a protective factor and pandemic burnout would function as a risk factor of mental health in a new wave of COVID-19 outbreak at Time 2. In particular, the following are examined:(1)A higher level of family resilience at Time 2 would be related to a lower level of anxiety and depression at Time 2;(2)A higher level of pandemic burnout at Time 2 would be related to a higher level of anxiety and depression at Time 2;(3)Family resilience and pandemic burnout at Time 2 would uniquely predict anxiety and depression at Time 2 over and above individual and family resilience at Time 1.

## 2. Materials and Methods

### 2.1. Participants

Participants from mainland China were recruited via an online survey platform called Credamo (www.credamo.com, accessed on 13 May 2022), which is a professional research data platform with survey distribution and data modeling services. The first and second waves of surveys were distributed in September 2021 and April 2022, respectively. On the first page of the survey, the purposes of the study were explained, and participants were invited to complete surveys at both times. Participants then indicated understanding of their rights and granted consent by pressing the “continue to next page”. Fifteen Chinese Yuan (CNY) were given to participants as compensation upon completion of each survey. Participants answered the questions anonymously, and only codes were used to match participants’ completed surveys at different time points. These responses are saved in an encrypted local drive and were accessible only to researchers of this project. Research ethics approval has been granted by the Human Research Ethics Committee of the research team’s affiliated university. 

There were 1135 participants recruited at Time 1 [46], and only 832 of them also completed the survey at Time 2. Attrition analyses showed that compared to those who dropped out after Time 1 (n = 303), those who also completed surveys at Time 2 (n = 832) were significantly older (M_difference_ = 1.32, *p* = 0.001), less anxious and depressive (M_difference_= −0.30, *p* = 0.012), and reported higher individual resilience (M_difference_= 0.86, *p* = 0.021) at Time 1. However, there was no significant difference in gender ratio, educational attainment, and family resilience at Time 1 between these two groups (p_s_ > 0.05).

Among participants who completed surveys at both times (n = 832), those who reported inconsistent demographic information, such as gender and year of marriage, were not included in subsequent analyses. Thus, the final analysis included 796 participants who responded at both time points. As shown in Table 1, the final sample was fairly gender-balanced and highly educated, had nearly all employed either full-time or part-time, and had more than half married.

### 2.2. Measures

#### 2.2.1. Anxiety and Depression 

Data for Time 2 were measured with the 4-item ultra-brief screening scale for anxiety and depression (PHQ-4; [47]). The items (e.g., “Over the last 2 weeks, how often have you been feeling down, depressed, or hopeless?”) were rated on a 4-point scale, ranging from 1 (less than usual) to 4 (much more than usual). The scores were summed, with a higher score reflecting a higher level of anxiety and depression. The Cronbach’s alpha was 0.73 in the current sample. 

#### 2.2.2. Pandemic Burnout 

Data for Time 2 were measured with the 10-item COVID-19 Burnout Scale (COVID-19-BS, [45]), which has been validated in a Chinese sample with satisfactory psychometric properties [44]. We have adjusted the wordings to tap on how much burnout people were feeling specifically toward the current COVID outbreak (e.g., “Under the current outbreak, how often do you feel tired when you think of COVID?”). Participants responded on a five-point scale ranging from 1 (never) to 5 (always). Responses were summed, with higher scores indicating a higher level of pandemic burnout. The internal consistency of this scale was high (α = 0.91). 

#### 2.2.3. Family Resilience

Data for Times 1 and 2 were measured with the 16-item Family Resilience Scale Short Form (FRS16; [46]), which has been validated in both the U.S. and Chinese samples. Participants indicated the extent to which the statements describe his/her family (e.g., “We can compromise when problems come up.”) on a 4-point scale (ranging from 1 = strongly disagree to 4 = strongly agree). The scale demonstrated satisfactory reliability at both Time 1 (α = 0.77) and Time 2 (α = 0.81).

#### 2.2.4. Individual Resilience 

Data for Time 1 were measured with the 10-item Connor-Davidson Resilience Scale (CD-RISC) [48], which taps on the participant’s individual resilience (e.g., “I am able to adapt when changes occur.”). Responses were made on a 5-point scale (ranging from 0 = not true at all to 4 = true nearly all the time). Scores were summed across items, where individuals with higher scores were more resilient. This scale had a Cronbach’s alpha of 0.90.

#### 2.2.5. Demographic Information 

Demographic information asked included age, educational attainment, marital status, number of children, employment, annual family income, and religion.

### 2.3. Data Analytic Strategy

As we are interested in whether Time 2 pandemic burnout and family resilience each uniquely predicts anxiety and depression at Time 2 and whether such associations are beyond the effects of individual and family resilience at Time 1, we examined these in sequence with a hierarchical regression model via SPSS 26. 

## 3. Results

Descriptive statistics of and bivariate correlations among key variables are summarized in Table 2. Results show that being younger, non-married, unemployed, childless, and having lower annual household income were associated with a higher level of depression and anxiety at Time 2. Both individual resilience and family resilience were negatively correlated with anxiety and depression. Family resilience was more strongly correlated with anxiety and depression at Time 2. Lastly, Time 2 pandemic burnout was positively correlated with Time 2 anxiety and depression and negatively correlated with family resilience at Times 1 and 2. 

To test our hypotheses, a hierarchical regression analysis was conducted to examine the main and interaction effects of Time 2 family resilience and pandemic burnout on anxiety and depression at Time 2, after considering effects of demographics and individual and family resilience at Time 1. Demographic information that was significantly related to anxiety and depression (either in current or previous studies), i.e., age, gender, marriage status, parenthood, employment status, and household income, was entered in Block 1; individual and family resilience at Time 1 were entered in Block 2; pandemic burnout and family resilience at Time 2 were entered in Block 3; and, lastly, the interaction term between family resilience and pandemic burnout at Time 2 was entered in Block 4.

Results of the regression are summarized in Table 3. All variables accounted for 55.3% of the variances in anxiety and depression at Time 2. Block 3 of the regression analysis indicated that family resilience and pandemic burnout at Time 2 showed significant incremental prediction of anxiety and depression at Time 2, even after controlling for effects of demographics and individual and family resilience at Time 1. Time 2 family resilience negatively and Time 2 pandemic burnout positively predicted anxiety and depression at Time 2; therefore, hypotheses 1 and 2 are supported. Hypothesis 3 is also supported. In Block 4, results showed that family resilience at Time 2 interacted with pandemic burnout to influence anxiety and depression at Time 2 (Figure 1). Post hoc analyses of the interaction effect showed that among participants who reported high pandemic burnout at Time 2, anxiety and depression at Time 2 was higher for those with low family resilience at Time 2 than those with high family resilience at Time 2 (means = 4.60 and 3.41, respectively; *p* < 0.01). However, among participants who reported low pandemic burnout at Time 2, there was no significant difference on anxiety and depression at Time 2 between participants with low and high family resilience (means = 1.89 and 1.80, respectively; *p* = 4.38). In other words, current family resilience mitigated the negative effect of high pandemic burnout on current anxiety and depression.

## 4. Discussion

The successive waves of COVID-19 outbreaks in the last three years have created a serious global environmental health threat. Countries have implemented varying preventative measures to lessen the risk of severe coronavirus cases and deaths, as well as to reduce the strain on healthcare systems. These have created significant changes in people’s daily lives. Research has consistently indicated negative mental health outcomes across different populations and countries as a result of the COVID-19 pandemic [2,3,4,5,6]. The present findings are largely consistent with available literature regarding demographic and individual factors that might affect mental health during the pandemic. Individuals who were younger, childless, unemployed, with lower household income, and with lower individual resilience were at a higher risk of anxiety and depression during the pandemic outbreak [49,50,51,52,53,54]. 

Previous research has suggested that crucial components of family processes, such as routines and dynamics, might be shaken in the face of the COVID-19 pandemic [55]. The present study found that family resilience showed moderate stability with a six-month interval, suggesting that family resilience might vary over time in response to environmental and life changes. The present study has also found that only family resilience at Time 2, not at Time 1, was negatively associated with individuals’ anxiety and depression at Time 2. Implication can be made that not only family resilience is transient and subjective to context, its positive influence on mental health is also bounded by time. Future research on family resilience should identify which of its components are more stable and which are more susceptible to changes (e.g., belief system vs. communication processes) in order to optimize intervention design.

Available literature has shown that family resilience was related to individual resilience among patients and caregivers [56,57]. The present study has also found this association among the general public. Furthermore, this study demonstrates that current family resilience had a main effect on current anxiety and depression, even after controlling demographic and individual resilience. These results are in line with available research [25] that found both individual and family resilience were uniquely and negatively associated with anxiety and depression. Furthermore, the present study reveals that current family resilience interacted with pandemic burnout to influence current mental health. In particular, the protective effect of family resilience on mental health was the most salient when individuals reported a high level of pandemic burnout. These findings have significant implications for prevention and intervention programs that aim to combat the mental health risk during the pandemic period. These programs should focus on cultivating or building family resilience, especially among individuals who reported high pandemic burnout.

The present study focuses on the unique predictive value of family resilience and pandemic burnout. Nevertheless, it is possible that mediation mechanisms exist among these variables. For instance, pandemic burnout might increase anxiety and depression levels by reducing family resilience; alternatively, family resilience might reduce anxiety and depression levels by lowering pandemic burnout. As the three variables were measured cross-sectionally, we cannot provide strong support for these mediation mechanisms in the current study. However, future investigation can further examine a mechanistic account through which family resilience and pandemic burnout influences depression and anxiety via each other with intervention studies in which family resilience is experimentally manipulated and multiple waves of outcome measures are included. 

The investigation of burnout during the COVID-19 pandemic has predominantly focused on service providers. To our knowledge, the only study that investigated the unique influence of pandemic burnout on anxiety and depression beyond individual resilience was conducted with a Polish general public sample [58]. Thus, our finding regarding the main effect of pandemic burnout and its interaction effect with family resilience on mental health indicates a cultural convergence of pandemic burnout as a risk factor of mental health. Our findings suggest that those who were younger, non-married, childless, unemployed, and of a lower income family were also more likely to feel burnout during new waves of COVID-19 outbreaks, whereas those with higher individual and family resilience showed lesser burnout. It implies that groups of people with these characteristics are more vulnerable to anxiety and depression compared to their counterparts. They might be facing more than one risk, wherein these factors might exacerbate each other, resulting in poorer mental health outcomes. 

Furthermore, a previous study found that when individuals held negative attitudes towards the preventative policy of their city, they felt more burnout towards the pandemic [59]. Hence, future studies are encouraged to investigate the degree to which and how different systems (e.g., organization, country, culture) shield individuals from or amplify the negative effects of the prolonged exposure to environmental deterioration and strict preventive measures in relation to the pandemic outbreak. 

There are some caveats of current study. Mainland China has applied the most stringent preventive policy of the world at the time of investigation [60]. However, the prevalence of new infected cases and preventive policies varied in different regions of China. Pandemic burnout might be perceived differently by participants who resided in different provinces of China. Secondly, self-selection bias could not be ruled out as resilient individuals with better mental health might be more likely to complete the surveys at both time points. Thirdly, mental health was assessed with only a short self-reported scale without clinical assessment. Family resilience was measured from participants only, without input from other family members. Lastly, there was no history of COVID-19 infection and vaccination of participants, which might affect the associations between various factors and mental health during the pandemic.

## 5. Conclusions

The successive waves of COVID-19 pandemic outbreaks and implementation of various preventative measures have persistent influences on our environment and mental health. This study explores psychosocial risk and protective factors of anxiety and depression during COVID-19 with a two-wave longitudinal study. Our findings reveal that family resilience and pandemic burnout had incremental influences beyond demographic factors and individual resilience. Among the general public, family resilience was a protective factor, whereas pandemic burnout was a risk factor of mental health. In particular, family resilience would mitigate the negative impact of high pandemic burnout on mental health. The present research has significant implications for designing prevention and intervention programs that aim at maintaining and/or enhancing mental well-being during the pandemic. For instance, both prevention and intervention could focus on family functioning in addition to individual aspects. Past studies suggested that family resilience could be enhanced by systematic intervention [61,62]. Future research could investigate the efficacy of such interventions in the Chinese population. Furthermore, the current COVID-19 pandemic has opened work-from-home options for many industries, resulting in changes to the work–family boundary and increased work–family interference [63,64]. This change may further complicate the interrelationship between family functioning and occupational functioning. Future research could have a bidirectional influence between family and occupational resilience during the pandemic and the conditions under which (e.g., the need for teleworking) their association is stronger.

## Figures and Tables

**Figure 1 ijerph-20-03803-f001:**
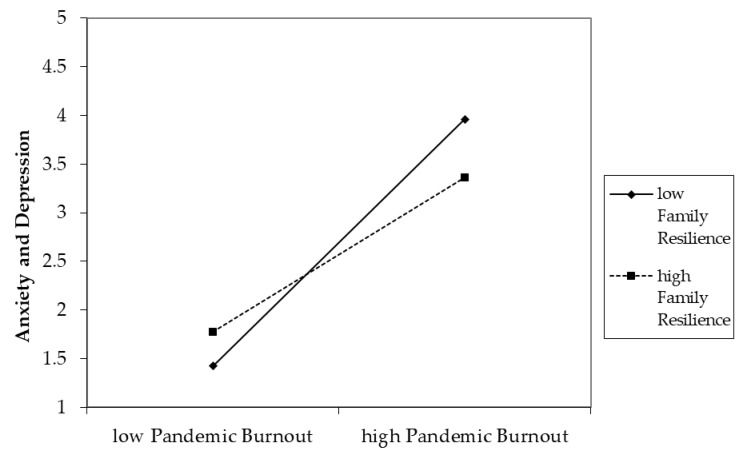
Simple slope analyses of the moderation effect of family resilience on the association between pandemic burnout and anxiety and depression at Time 2. Note all variables were measured at Time 2.

**Table 1 ijerph-20-03803-t001:** Sample Characteristics.

		*n*	%	*M* (*SD*)	Range
1	Gender				
	Female	454	57		
	Male	342	43		
2	Age (in years)			30.26 (6.03)	18–56
3	Annual household income (USD)				
	>1069	14	1.8		
	1069–2285	64	8.0		
	2285–3626	94	11.8		
	3626–5682	58	7.3		
	5682–11,065	68	8.5		
	<11,066	498	62.6		
4	Employment status				
	Full-time/part-time	790	99.2		
	Unemployed	5	0.6		
	Retired	1	0.1		
5	Average weekly working hours			41.62 (10.95)	0–120
6	Relationship status				
	Single	152	19.1		
	Married	562	70.6		
	In relationship	82	10.3		
7	Year-in marriage	562		6.74 (4.65)	1–30
8	No. of participants with child	539	67.7		
9	Educational level				
	Secondary school or below	31	3.9		
	Associate/Diploma	80	10.1		
	Bachelor’s degree	594	74.6		
	Postgraduate degree	91	11.4		
10	Religion				
	Christian	17	2.1		
	Catholic	1	0.1		
	Buddhist	82	10.3		
	Atheist	691	86.8		
	Other	5	0.6		

**Table 2 ijerph-20-03803-t002:** Descriptive statistics and bivariate correlations of key variables (*n* = 796).

	Variable	α	M	SD	Min	Max	1	2	3	4	5	6	7	8	9	10
1	Age	-	30.26	6.03	18	56										
2	Gender ^a^	-	-	-	-	-	−0.104 **									
3	Marital status ^b^	-	-	-	-	-	0.597 ***	−0 0.042								
4	Having child ^c^	-	-	-	-	-	0.581 ***	−0 0.062	0.923 ***							
5	Employment status ^d^	-	-	-	-	-	−00.023	−0.024	−0.051	−0.044						
6	Annual family income	-	-	-	-	-	0.041	−0.052	0.105 **	0.099 **	−0.097 **					
7	High anxiety and depression (T2)	0.733	2.71	1.94	0	12	−0.227 ***	0.039	−0.319 ***	−0.322 ***	0.117 **	−0.108 **				
8	High individual resilience (T1)	0.902	29.98	5.71	5	40	0.227 ***	−0.045	0.362 ***	0.363 ***	−0.101 **	0.176 ***	−0.455 ***			
9	Family resilience (T1)	0.769	49.42	4.71	24	62	0.166 ***	−0.015	0.343 ***	0.350 ***	−0.068	0.175 ***	−0.296 ***	0.625 ***		
10	Family resilience (T2)	0.805	51.55	5.40	22	62	0.128 ***	0.010	0.297 ***	0.275 ***	−0.105 **	0.209 ***	−0.408 ***	0.558 ***	0.610 ***	
11	Pandemic burnout (T2)	0.906	22.06	7.27	10	47	−0.229 ***	0.041	−0.304 ***	−0.305 ***	0.089 *	−0.141 ***	0.672 ***	−0.350 ***	−0.249 ***	−0.335 ***

Notes: *** *p* < 0.001; ** *p* < 0.01; * *p* < 0.05; ^a^ gender, 1 = male, 2 = female; ^b^ marital status, 1 = non-married, 2 = married; ^c^ having child, 0 = no, 1 = yes; ^d^ employment status, 1 = employed, 2 = unemployed/retired.

**Table 3 ijerph-20-03803-t003:** Hierarchical linear regression model on the effects of demographics, individual and family resilience, and pandemic burnout on anxiety and depression (*n* = 796).

Block	Predictor	F	∆F	R^2^	∆R^2^	β	t	*p*
1		18.498 ***	18.498 ***	0.123	0.123			
	Age					−0.050	−1.187	0.236
	Gender ^a^					0.017	0.508	0.612
	Marital status ^b^					−0.120	−1.355	0.176
	Having child ^c^					−0.170	−1.958	0.051
	Employment status ^d^					0.096	2.858	0.004
	Annual household income					−0.067	−1.974	0.049
2		31.600 ***	62.285 ***	0.243	0.120			
	Age					−0.044	−1.128	0.260
	Gender ^a^					0.009	0.293	0.769
	Marital status ^b^					−0.059	−0.715	0.475
	Having child ^c^					−0.102	−1.250	0.212
	Employment status ^d^					0.068	2.189	0.029
	Annual household income					−0.018	−0.559	0.576
	T1 individual resilience					−0.387	−9.468	0.000
	T1 family resilience					0.017	0.409	0.682
3		88.129 ***	230.521 ***	0.523	0.280			
	Age					−0.014	−0.438	0.661
	Gender ^a^					0.007	0.286	0.775
	Marital status ^b^					0.007	0.100	0.920
	Having child ^c^					−0.065	−1.008	0.314
	Employment status ^d^					0.037	1.498	0.135
	Annual household income					0.032	1.269	0.205
	T1 individual resilience					−0.211	−6.177	<0.001
	T1 family resilience					0.077	2.190	0.029
	T2 family resilience					−0.138	−4.129	<0.001
	T2 pandemic burnout					0.551	20.081	<0.001
4		89.154 ***	52.213 ***	0.553	0.030			
	Age					−0.008	−0.259	0.795
	Gender ^a^					0.001	.045	0.964
	Marital status ^b^					0.003	.046	0.964
	Having child ^c^					−0.065	−1.042	0.298
	Employment status ^d^					0.034	1.395	0.163
	Annual household income					0.025	1.015	0.310
	T1 individual resilience					−0.240	−7.215	<0.001
	T1 family resilience					0.056	1.650	0.099
	T2 family resilience					−0.035	−0.978	0.328
	T2 pandemic burnout					0.522	19.407	<0.001
	T2 pandemic burnout x T2 family resilience					−0.196	−7.226	< 0.001

Notes: ^a^ gender, 1 = male, 2 = female; ^b^ marital status, 1 = non-married, 2 = married; ^c^ having child. 0 = no, 1 = yes; ^d^ employment status, 1 = employed, 2 = unemployed/retired. *** *p* < 0.001, ∆F = change in F statistic across blocks, R2 = variance explained by each model, ∆R2 = additional variance explained compared to previous model.

## Data Availability

The data supporting the conclusions of this article will be made available by the authors, without undue reservation.

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
