# Peer review of "The Role of Family Resilience and Pandemic Burnout on Mental Health: A Two-Wave Study in China"

_ijerph, 2023, doi:10.3390/ijerph20053803_

Round 1

Reviewer 1 Report

Thank you for giving me an opportunity to revise enclosed manuscript. It deals with nowadays crucial issue which mental health and pandemic burnout definitely is. Adding family resilience made it even more interesting both for scholars and policy makers.

Theoretical background has been well-described, as well as justification for the choice of study period. Moreover, discussion section is relevant and includes very important notes about the study and similar efforts from the others scholars.

I'd like to underline that Authors has well-pointed out limtations of the study, especially with the sample.

Just several doubts during reading this manuscript:

1. Between the lines 100-104 Authors has written i.e. : "(..) the shadow of COVID-19 is still casting all over our everyday life with little signs of cessation. (...)". Is the language used in those lines proper for scientific text? Being frank, I have some concerns. Please, rethink the style there.

2. Is it appropriate to support H1 nad H2 using only bivariate correlation? It's a very simple measure.

3. Please expand the Conclussion section, mainly focusing on recommendations.

Author Response

Thank you for your careful review, and the opportunity for revision of the manuscript “The Role of Family Resilience and Pandemic Burnout on Mental Health: A Two-wave Study in China”, (manuscript ID: ijerph-2189527).

The followings are our replies to reviewers’ comments. Besides addressing specific comments and concerns, we also made changes that we thought could further improve the clarity and accuracy of scientific expression. All page numbers refer to revised manuscript in this resubmission. 

---------------------------------------------------------------------------------

We quote your concerns and address each in turn. The page number below refers to the revised manuscript.

Reviewer 1’s comments

  1. “Between the lines 100-104 authors have written i.e.: "(..) the shadow of COVID-19 is still casting all over our everyday life with little signs of cessation. (...)". Is the language used in those lines proper for scientific text? Being frank, I have some concerns. Please, rethink the style there.”

Reply:

We have adjusted the framing with more straightforward expression (lines 102-103, p.3).

---------------------------------------------

  1. “Is it appropriate to support H1 and H2 using only bivariate correlation? It's a very simple measure.”

            Reply:

            We use regression to support H1 and H2 instead, revision is made on lines 244-246 (p.8).

---------------------------------------------

  1. “Please expand the Conclusion section, mainly focusing on recommendations.”

            Reply:

Recommendations are added in the conclusion section (lines 349-358, p.11). In particular, we have discussed the importance of studying family resilience intervention and the association between work and family resilience during the pandemic.

---------------------------------------------------------------------------------

Reviewer 2 Report

Dear Authors,

It was a pleasure to review your work. Overall, this is interesting research, not only because it focus on a very relevant area of intervention for professionals, and also it was developed in a specific period (COVID-19 lockdown). It is well developed, and it was a pleasure for me to review it.

Some small considerations to improve the work presented:

1. Page 4, line 160 - before using an abbreviation, indicate its name in full.

2. I felt the need of a sub-chapter describing data analyses procedures.

Page 6, line 214 - "moderate consistency" - I do not understand the expression in regard to the analysis that is being made.

3. Table 2 (page 7): since the researchers are using the mean sums, it can be relevant for the readers to have access in the table the amplitude (min and max) of the variables.

4. The regressions could be also discussed in respect to possible mediation effects.

5. I would remove result numbers from the discussion.

Congratulations for the work presented.

Best regards

Author Response

Thank you for your careful review, and the opportunity for revision of the manuscript “The Role of Family Resilience and Pandemic Burnout on Mental Health: A Two-wave Study in China”, (manuscript ID: ijerph-2189527).

The followings are our replies to reviewers’ comments. Besides addressing specific comments and concerns, we also made changes that we thought could further improve the clarity and accuracy of scientific expression. All page numbers refer to revised manuscript in this resubmission. 

---------------------------------------------------------------------------------

We quote your concerns and address each in turn. The page number below refers to the revised manuscript.

Reviewer 2’s comments

  1. “Page 4, line 160 - before using an abbreviation, indicate its name in full.”

            Reply:

            It has been added in line 161 (p.4).

----------------------------------------

  1. “I felt the need of a sub-chapter describing data analyses procedures.”

“Page 6, line 214 - "moderate consistency" - I do not understand the expression in regard to the analysis that is being made.”

Reply:

A sub-chapter describing data analytic strategy has been added in lines 213 to 217 (p.6).

The expression has been removed.

-----------------------------------------

  1. “Table 2 (page 7): since the researchers are using the mean sums, it can be relevant for the readers to have access in the table the amplitude (min and max) of the variables.”

            Reply:

            The minimal and maximal values of all key continuous variables were included in Table 2. As gender, marital status, whether having child, employment status and annual family income are categorical, therefore, the minimal and maximal values are not available or not meaningful to present. Instead, the percentage and count of each category were presented in Table 1 (p.5) for these variables.

-----------------------------------------

  1. “The regressions could be also discussed in respect to possible mediation effects.”

Reply:

A discussion on possible mediation effects is added in lines 298 to 307 (p.11).

-----------------------------------------

  1. “I would remove result numbers from the discussion.”

Reply:

            The number in the result section has been removed (line 276, p.10).